# Anti-PD-1 immunotherapy leads to tuberculosis reactivation via dysregulation of TNF-α

Liku B Tezera[1,2]\*, Magdalena K Bielecka[1], Paul Ogongo[3,4], Naomi F Walker[5,6,7], Matthew Ellis[8], Diana J Garay-Baquero[1,2], Kristian Thomas[1], Michaela T Reichmann[1], David A Johnston[1], Katalin Andrea Wilkinson[9], Mohamed Ahmed[3], Sanjay Jogai[1], Suwan N Jayasinghe[10], Robert J Wilkinson[9,11], Salah Mansour[1,2], Gareth J Thomas[8], Christian H Ottensmeier[8], Alasdair Leslie[3,12], Paul T Elkington[1,2]

[1]NIHR Biomedical Research Centre, School of Clinical and Experimental Sciences, Faculty of Medicine, University of Southampton, Southampton, United Kingdom; [2]Institute for Life Sciences, University of Southampton, Southampton, United Kingdom; [3]Africa Health Research Institute, KwaZulu Natal, South Africa; [4]Department of Tropical and Infectious Diseases, Institute of Primate Research, National Museums of Kenya, Nairobi, Kenya; [5]Wellcome Centre for Infectious Diseases Research in Africa, Institute of Infectious Disease and Molecular Medicine, University of Cape Town, Cape Town, South Africa; [6]TB Centre and Department of Clinical Research, London School of Hygiene and Tropical Medicine, London, United Kingdom; [7]Department of Clinical Sciences, Liverpool School of Tropical Medicine, Liverpool, United Kingdom; [8]NIHR Biomedical Research Centre, School of Cancer Sciences, University of Southampton, Southampton, United Kingdom; [9]The Francis Crick Institute, London, United Kingdom; [10]BioPhysics Group, Department of Mechanical Engineering, University College London, London, United Kingdom; [11]Department of Infectious Diseases, Imperial College London, London, United Kingdom; [12]Department of Infection and Immunity, University College London, London, United Kingdom

\*For correspondence:
l.tezera@soton.ac.uk

**Abstract** Previously, we developed a 3-dimensional cell culture model of human tuberculosis (TB) and demonstrated its potential to interrogate the host-pathogen interaction (Tezera et al., 2017a). Here, we use the model to investigate mechanisms whereby immune checkpoint therapy for cancer paradoxically activates TB infection. In patients, PD-1 is expressed in *Mycobacterium tuberculosis* (Mtb)-infected lung tissue but is absent in areas of immunopathology. In the microsphere model, PD-1 ligands are up-regulated by infection, and the PD-1/PD-L1 axis is further induced by hypoxia. Inhibition of PD-1 signalling increases Mtb growth, and augments cytokine secretion. TNF-α is responsible for accelerated Mtb growth, and TNF-α neutralisation reverses augmented Mtb growth caused by anti-PD-1 treatment. In human TB, pulmonary TNF-α immunoreactivity is increased and circulating PD-1 expression negatively correlates with sputum TNF-α concentrations. Together, our findings demonstrate that PD-1 regulates the immune response in TB, and inhibition of PD-1 accelerates Mtb growth via excessive TNF-α secretion.

## Introduction

Tuberculosis (TB) continues to be a global health pandemic, killing more people than any other infection (*Wallis et al., 2016*). TB involves a complex host-pathogen interaction, with humans and *Mycobacterium tuberculosis* (Mtb) having undergone prolonged co-evolution (*Menardo et al., 2019*). TB has often been thought to primarily result from loss of immune control, because approximately 90% individuals infected with TB never progress to active disease, and this progression is increased in the context of immune deficiency; such as in cases of HIV infection, in infants, people with genetic deficiency of the IL-12/IFN-γ signalling pathway or after anti-TNF-α antibody treatment (*O'Garra et al., 2013*). However, an emerging concept is that an excessive immune response to Mtb may be equally harmful. Standard disease paradigms predict that immune activation resulting from the administration of checkpoint inhibitors should lead to better control of Mtb infection (*Zumla et al., 2016*). However, counter-intuitively these agents seem to be activating TB, as evidenced by recent reports of TB developing in patients treated for malignancy with immune checkpoint inhibition, often rapidly after commencing therapy (*Fujita et al., 2016*; *Lee et al., 2016*; *Chu et al., 2017*; *Picchi et al., 2018*; *Jensen et al., 2018*; *Elkington et al., 2018*; *He et al., 2018*; *Takata et al., 2019*; *Barber et al., 2019*; *Tsai et al., 2019*; *van Eeden et al., 2019*). Consistent with this emerging clinical phenomenon, programmed death (PD-1) deficient mice are highly susceptible to TB, dying more rapidly than T-cell deficient mice (*Lázár-Molnár et al., 2010*; *Barber et al., 2011*).

PD-1 and its ligand PD-L1 are expressed in human granulomas (*Elkington et al., 2018*), suggesting a regulatory role at the site of disease. TB granulomas are hypoxic (*Belton et al., 2016*), and PD-L1 is up-regulated by hypoxia (*Noman et al., 2014*), further suggesting a mechanistic link between hypoxia and the PD-1/PD-L1 axis within TB lesions. In this study, we investigate the expression patterns of PD-1 and PDL-1 within TB infected human lung tissue and the relationship between PD-1 and anti-TB immunity. Next, using a human 3D cell culture model of TB (*Tezera et al., 2017a*), we show that hypoxia increases expression of PD-1 and its ligands, that PD-1 inhibition increases Mtb growth. Surprisingly, TNF-α is primarily responsible for this effect, and TNF-α neutralisation reverses the anti-PD-1 induced phenotype.

## Results

### PD-1 is expressed in human TB granulomas but not in areas of immunopathology

First, we investigated the presence and localisation of PD-1-expressing T cells in human pulmonary TB. We hypothesised that PD-1 would be expressed by T cells in the lung of patients with TB, and at a higher frequency than in the blood. In thirty-five patients undergoing medically indicated lung resection to treat TB or TB sequalae, PD-1 expression was measured on T cells isolated from the lung and matched blood samples, available for 23 patients, by flow cytometry. Overall, PD-1 expression in homogenized lung tissue was highly variable, with a trend towards increased PD-1 in both CD4 and CD8 T-cells from the lung compared to matched blood, which reached statistical significance for CD8 T-cells (*Figure 1A*). Lung tissue from healthy individuals was not available for study, however, the median frequency of 11% and 14% of PD-1+ CD4 and CD8 T-cells observed are generally lower than recently reported for healthy human lung tissue from organ donors of approximately 50% for both cell types (*Snyder et al., 2019*). As lung tissue is highly perfused with blood, distinguishing cells of lung or blood origin is challenging.

To explore this further, we stained lung samples for canonical markers of tissue resident T-cells, CD69 and CD103 (*Snyder et al., 2019*). PD-1 expression on lung CD4 cells was predominantly restricted CD69+ T-cells, with a smaller proportion also expressing CD103 (*Figure 1B*, overall frequencies *Supplementary file 1*), consistent with a tissue resident phenotype and in contrast to PD-1 expressing cells in blood, which are largely CD69 negative (*Figure 1—figure supplement 1*). Lung CD8 T-cells were also found to predominantly express CD69 in addition to CD103, again in contrast to CD8 T-cells in circulation. Therefore, these data are consistent with expression of PD-1 on lung

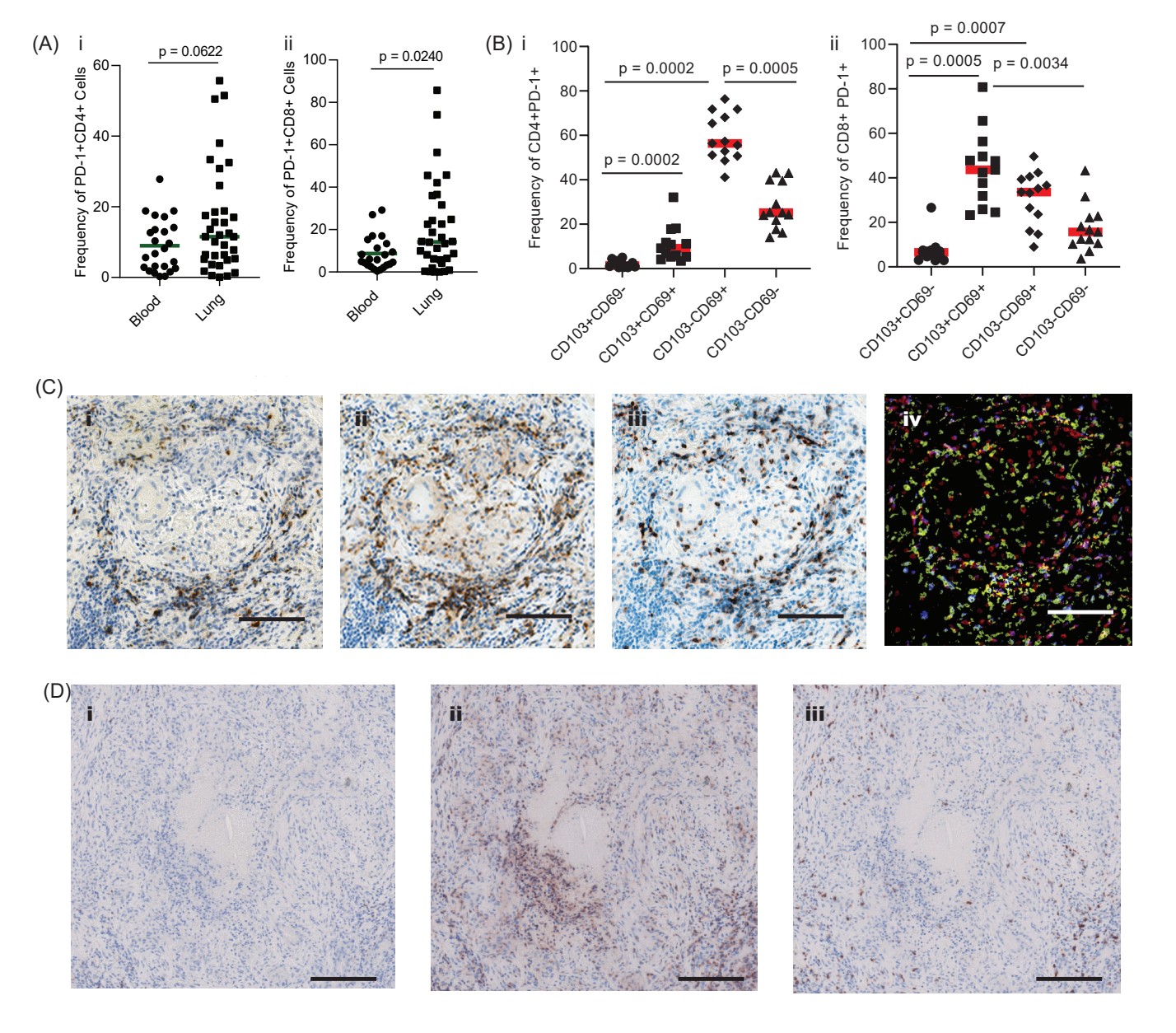

**Figure 1.** PD-1 is expressed in human TB granulomas. (A) Analysis of PD-1 expression by T cells in the lung and peripheral circulation of thirty-five TB patients undergoing medically indicated lung resection. PD-1 shows a trend towards higher expression by lung CD4[+] (i) and is significantly higher on lung CD8[+] (ii) T cells. Significance analysed by one-tailed unpaired Mann-Whitney test. (B) Flow cytometric analysis of lung parenchyma CD4[+] (i) and CD8[+] (ii) T cells from TB patients based on the expression of PD-1, CD69 and CD103 demonstrates increased PD-1 expression in the resident T cells in the lung parenchymal cells. Significance analysed by Kruskal-Wallis test with Dunn's multiple comparison test. (C) Immunohistochemical staining for PD-1[+], CD4[+] and CD8[+] expression in human lung TB granulomas. PD-1 is expressed around the central macrophage core in the same region as CD4[+] (ii) and CD8[+] (iii) T cells. Co-localization of PD-1 (blue), CD4[+] (red) and CD8[+] (yellow) using false colour of the immunostaining shows co-localisation of PD-1 with both CD4[+] and CD8[+] cells (purple and green respectively) (iv). Scale bar 100 μm. (D) PD-1 is not expressed in caseating granulomas where immunopathology is present in human lung biopsies (i). Six biopsies taken as part of routine clinical care were studied. CD4[+] (ii) and CD8[+] (iii) expressing cells are present in the same area, and so absence of PD-1 immunoreactivity is not due to lack of viable cells. Scale bar 200 μm.

The online version of this article includes the following figure supplement(s) for figure 1:

**Figure supplement 1.** PD-1 expression on peripheral CD4[+] and CD8[+] T-cells is predominantly on CD103- and CD69-negative cells.

**Figure supplement 2.** PD-1 expressing cells are absent in the immediate region surrounding caseous necrosis in human TB granulomas.

tissue resident T-cells in patients with active or previous pulmonary TB infection. However, TB immunopathology is highly heterogenous within the lung and these analyses do not provide information as to the localization of PD-1 expression within TB lesions.

To address this question, we performed immunohistochemical analysis of human lung biopsies of patients with active TB. Within stable granulomas with an intact cellular structure, PD-1 expression was present around the central macrophage core (*Figure 1C–i*). Both CD4 and CD8 T cells were present in the granuloma (*Figure 1C–ii* and -iii), and co-localisation analysis demonstrated that a proportion of both T cell subtypes expressed PD-1 (*Figure 1C*-iv), consistent with the flow cytometry data. In contrast, in caseating granulomas with evident immunopathology, PD-1 expression was totally absent (*Figure 1D–i* and *Figure 1—figure supplement 2*). The lack of immunoreactivity was not due to lack of viable cells, as CD4 and CD8 expressing T cells were present within the same area (*Figure 1D–ii* and -iii). Therefore, in human granulomas, PD-1 is expressed in areas where the host-pathogen interaction appears stable but is absent in regions of immunopathology.

## The PD-1/PD-L1 axis is up-regulated in the 3D microsphere model

Taken together, these data from Mtb infected patients demonstrate that PD-1 is expressed by lung resident T-cells and may be required to prevent destructive lung disease. However, whether this is a causal associated remains unclear. We therefore explored the biological role of PD-1 expression in a 3-dimensional (3D) human tissue culture model. TB is a human disease characterised by a prolonged host-pathogen interaction in 3D, and is regulated by the extracellular matrix, and previously we have previously developed a 3D cell culture model incorporating extracellular matrix to investigate TB pathogenesis (*Tezera et al., 2017a*; *Tezera et al., 2017b*). Here, we first studied migration of cells within the alginate-collagen matrix by performing time-lapse microscopy (*Figure 2A* and *Video 1*). For this experiment only, UV killed Mtb was used to permit the imaging outside the containment level three laboratory. From 24 hr, progressive accumulation occurs around a central infected core, resulting in a large multicellular granuloma by 48 hr with dynamic cellular movement similar to the T cell mobility observed in mouse BCG granulomas (*Egen et al., 2008*).

We next investigated whether the PD-1/PD-L1 axis was up-regulated by infection in this 3D model, and found that Mtb infection increased PD-L1 and PD-L2 gene expression at 4 days post infection (*Figure 2B*). Furthermore, a hypoxic environment (1% oxygen) further increased PD-L2 expression. PD-1 gene expression was not increased at 72 hr by Mtb infection under normoxic conditions, but was up-regulated when infected cells were incubated in hypoxia (*Figure 2B*). The change in expression required both infection and hypoxia, as hypoxia did not increase PD-1 or PD-L1/2 expression alone (*Figure 2—figure supplement 1*). We then studied PD-1/PD-L1 cellular surface expression in microspheres by flow cytometry after one week of infection. Mtb infection slightly increased PD-1 surface expression on CD4$^+$ T cells in normoxia, which was augmented by hypoxia (*Figure 2C*). Similarly, Mtb infection increased PD-1 expression on CD8$^+$ T cells, which was greater in hypoxia (*Figure 2D*). Furthermore, on CD14$^+$ CD11b$^+$ cells, PD-L1 expression was increased by infection and more so in hypoxia (*Figure 2E*). To investigate whether the effect of hypoxia on increased PD-1/PD-L1 expression was due to altered Mtb growth, we studied proliferation using Mtb expressing firefly luciferase and also colony counting on Middlebrook 7H11 agar. Hypoxia inhibited Mtb growth by both readouts (*Figure 2F* and *Figure 2—figure supplement 2A*). As expected, hypoxia also increased expression of the hypoxia inducible factor, HIF-1α, in host cells (*Figure 2G*), but did not have a significant effect on host cell survival (*Figure 2—figure supplement 2B*). Therefore, the increased PD-1 and PD-L1/2 expression in hypoxia was not due to increased Mtb growth or a change in host cell viability.

## PD-1 pathway inhibition increases Mtb growth in the 3D model

Having demonstrated that the PD-1/PD-L1 axis was up-regulated by infection in the 3D model, we investigated whether inhibition of this interaction modulated host control of Mtb. We initially studied chemical inhibitors of the PD-1/PD-L1 axis (Inhibitor 1, $C_{29}H_{33}NO_5$) and found that blocking PD-1/PD-L1 binding increased Mtb growth in a dose-dependent manner (*Figure 3A*). This effect occurred in both normoxia and hypoxia (*Figure 3—figure supplement 1*). No effect of PD-1 inhibition on cell survival was noted at day 7 or day 14, utilising two different assays suited to each time point (*Figure 3B and C*). Next, we studied spartalizumab, a humanised monoclonal antibody with high-

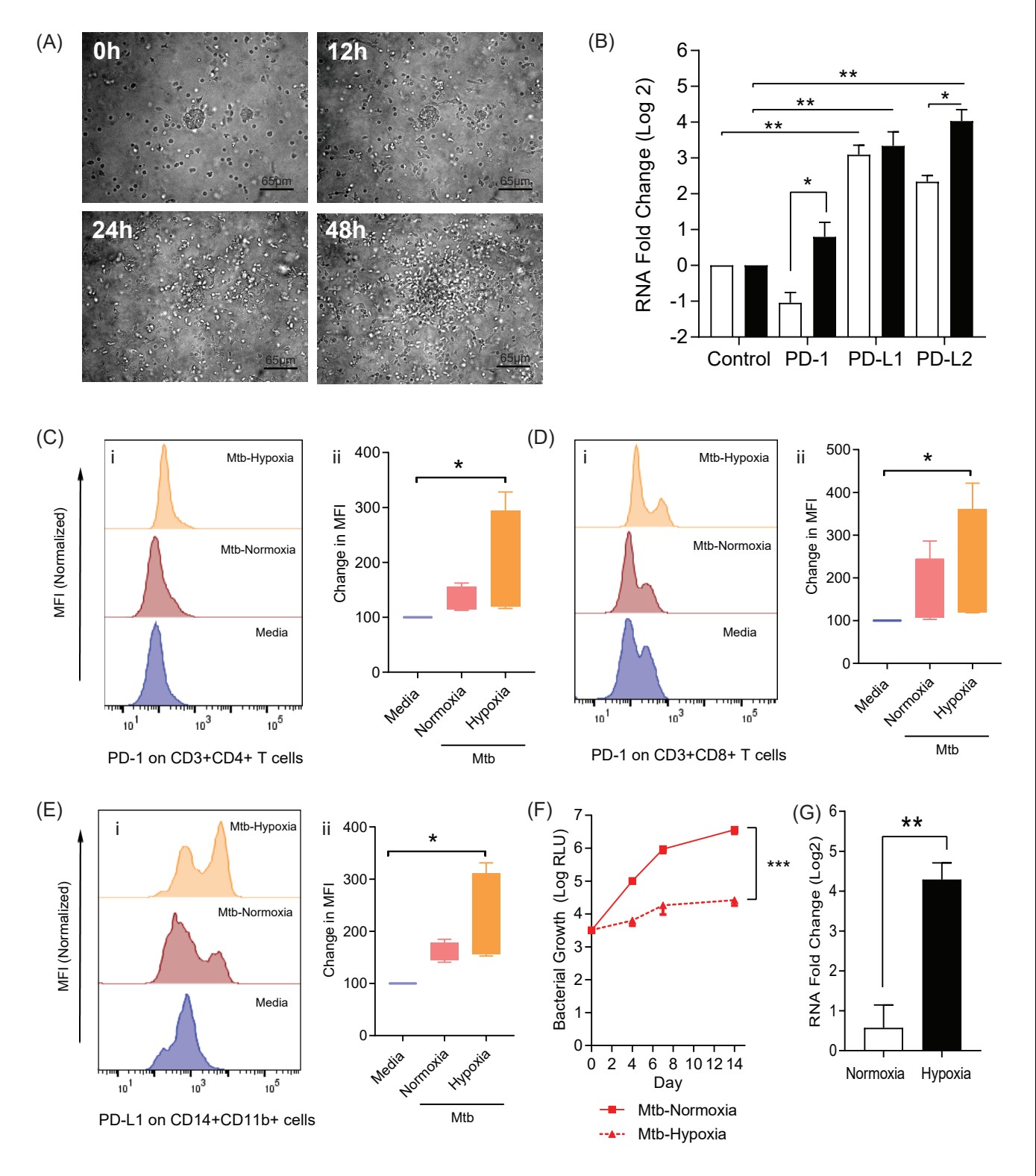

**Figure 2.** The PD-1/PD-L1 axis is upregulated in the 3D TB granuloma system. (**A**) Still images from time-lapse microscopy imaging demonstrating increasing cellular aggregation of PBMC around a focus of ultraviolet killed Mtb H37Rv in the 3D granuloma system at times 0, 12, 24 and 48 hr post encapsulation in the matrix. The Z projection shows the cells contained within the designated volume in a 2D reconstruction. Full time course in *Video 1*. (**B**) Gene expression of PD-1 and its ligands in the 3D microsphere model. RNA was extracted from live Mtb-infected PBMC and relative
*Figure 2 continued on next page*

Figure 2 continued

expression investigated by qRT-PCR at day four post infection. Open bars, normoxia, filled bars 1% hypoxia. PD-L1 and PD-L2 are upregulated by Mtb infection, and in 1% hypoxia PD-1 expression is increased and PD-L2 expression further augmented (n = 4). Results are normalised against the housekeeping genes GAPDH, β-Microbulin and FANTA and showed similar results. β-microglobulin used for (B). *p<0.05, **p<0.01. (C–E) Surface expression of PD-1 and PD-L1. PBMCs were decapsulated from Mtb-infected microspheres at day seven and surface expression of PD-1 and its ligand PD-L1 were analysed by flow cytometry. PD-1 is expressed in CD4+ (C) and CD8+(D) T cells in PBMC from Mtb infected microspheres incubated in normoxia. PD-1 expression was significantly upregulated in 1% hypoxia. Representative flow cytometry plots and level of expression of PD-1 by the CD4+ and CD8+ T cell fractions are shown (n = 4). (E) PD-L1 expression on CD14+CD11b+ cells within PBMC in Mtb infected microspheres is upregulated in both normoxia and 1% hypoxia at day 7 (n = 4). Significance of *p<0.05. (F) Growth of Mtb H37Rv ffLux+ in microspheres in normoxia and 1% hypoxia measured at day 3, 7 and 14. Hypoxia reduces Mtb growth. (G) Hypoxia inducible factor 1α (HIF-1α) mRNA levels were increased in Mtb-infected microspheres incubated in 1% hypoxia. RNA was extracted from decapsulated microspheres and normalised to uninfected microspheres in the same environment. Results were normalised to the housekeeping genes GAPDH, β-microglobulin and FANTA to check the housekeeping gene are not affected by hypoxia. Similar results all three of the housekeeping genes. β-microglobulin used for this graph. Significance ***p<0.001.

The online version of this article includes the following figure supplement(s) for figure 2:

**Figure supplement 1.** Hypoxia alone has no significant effect on expression of the PD-1/PD-L1/2 axis.
**Figure supplement 2.** Mtb growth is reduced in hypoxia, while host cell viability is unchanged.

affinity to PD-1 that blocks the interaction with PD-L1 and PD-L2 (*Kaplon and Reichert, 2019*). Consistent with the chemical inhibition, PD-1 pathway inhibition using spartalizumab increased Mtb growth in normoxia in a dose-dependent manner (*Figure 3D*). Similarly, in hypoxia spartalizumab increased Mtb growth (*Figure 3E*). Microspheres must be restored to normoxia to measure Mtb luminescence, and therefore a single time point was analysed. Anti-PD-1 antibody treatment had no significant effect on cell survival in either normoxia or hypoxia (*Figure 3F*).

## PD-1 inhibition increases secretion of multiple cytokines and growth factors

To investigate the underlying mechanism whereby PD-1 inhibition leads to increased Mtb growth, we studied secretion of cytokines, chemokines and growth factors by measuring accumulation in the media around Mtb-infected microspheres by Luminex array. Mtb infection increased secretion of numerous analytes (*Figure 4—figure supplement 1*), and inhibition of PD-1/PD-L1 signalling further significantly augmented secretion of twelve analytes compared to Mtb infection alone (*Figure 4*). A similar augmentation of analyte secretion was observed in hypoxic microspheres (*Figure 4—figure supplement 2*). The twelve analytes that were further increased above Mtb infection with concurrent PD-1 inhibition were IL-4, IL-6, IL-10, IL-12, TNF-α, IL-1RA, MIP-1α, MIP-1β, RANTES, G-CSF, GM-CSF and VEGF (individual concentrations, *Figure 4—figure supplement 1*). Anti-PD-1 alone increased TNF-α secretion, although to a lesser effect than in combination with Mtb infection (*Figure 4—figure supplement 3*).

## Exogenous TNF-α increases Mtb growth in microspheres

To establish which of these factors might be associated with increased Mtb growth, we added the significantly upregulated analytes either individually or in combination pools to 3D microspheres in normoxia at 'low' (*Figure 5*) and 'high' concentrations (*Figure 5—figure supplement 1*), as determined by the concentration measured in the secretion analysis. TNF-α was the dominant cytokine that increased Mtb growth, with other cytokines only having a minor effect (*Figure 5A*). Chemokines (RANTES, MIP-1α and MIP-1β) and growth factors (G-CSF) had no significant effect, while GM-CSF significantly

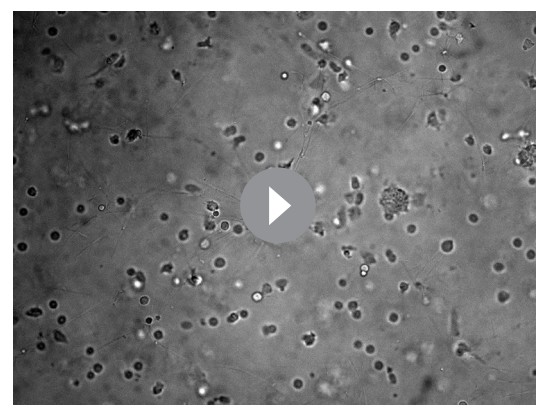

**Video 1.** Cell migration over 48 hr around a central cluster of macrophages infected with UV-killed Mtb within a 3D alginate-collagen matrix. Migration is seen in the first 24 hr, without aggregation, and then progressive granuloma formation occurs.
https://elifesciences.org/articles/52668#video1

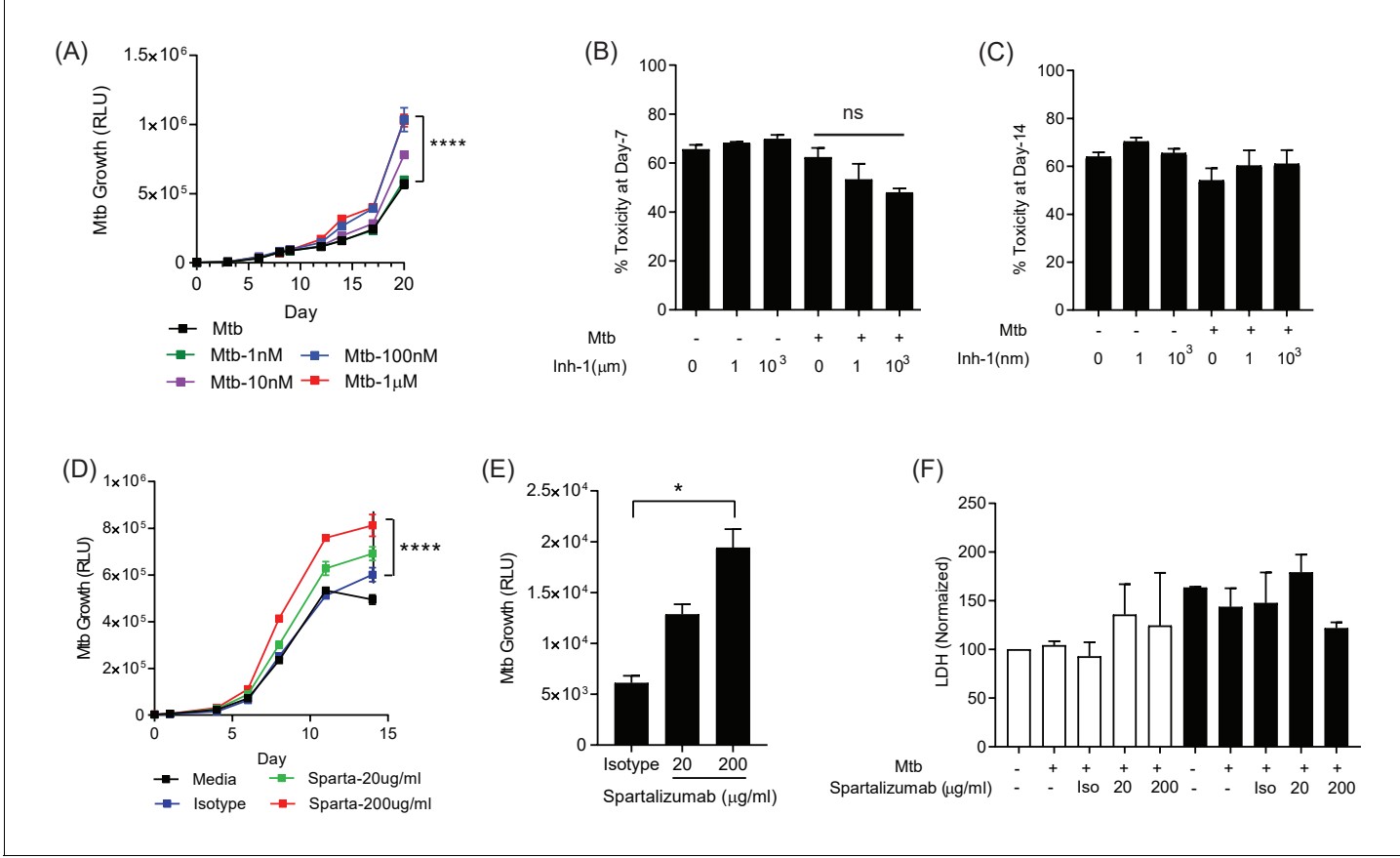

**Figure 3.** PD-1 pathway inhibition increases Mtb growth. (A) Inhibition of PD-1 receptors by small chemical inhibitor one increases Mtb growth in a dose-response manner (1–1000 nM). Inhibitor concentration 1 nM (green), 10 nM (purple), 100 nM (blue) and 1 μM (red). (B) Inhibitor one was not toxic to Mtb-infected PBMC, analysed by CytoTox-Glo assay (Day 7). (C) Cellular toxicity was no different at day 14 as analysed by LDH release. Concentration 1 and 1000 nM were analysed for toxicity. (D) Spartalizumab, a therapeutic monoclonal anti-PD-1 antibody, progressively increased Mtb growth in microspheres in normoxia in a dose-dependent manner. (E) Spartalizumab also increased Mtb growth in hypoxia. Media (black), isotype (blue), spartalizumab 20 μg/ml (green) and 200 μg/ml (red). (F) The anti-PD-1 antibody had no effect on cell survival in microspheres in normoxia (clear bars) and 1% hypoxia (filled bars). Cytotoxity is determined by measuring LDH release at day 14 and normalized by the control. ****p<0.0001.
The online version of this article includes the following figure supplement(s) for figure 3:

**Figure supplement 1.** Small chemical inhibition of PD-1/PD-L1 interaction in 1% hypoxia measured at day 14 shows a dose-dependent increase in Mtb growth with PD-1 inhibition.

increasing Mtb growth, but the effect was smaller than for TNF-α (*Figure 5B*). Additionally, the only cytokine combination that had a significant effect on Mtb growth was the pro-inflammatory pool containing TNF-α, IL-6 and IL-12 (*Figure 5C*). The addition of $Th_2$ cytokines, chemokines or other growth factors had no significant effect (individual growth curves at 'high' concentration, *Figure 5—figure supplement 1*). Furthermore, TNF-α had a progressive dose-dependent effect increasing Mtb growth within microspheres (*Figure 5D*).

To explore the effect of TNF-α further, we then generated microspheres incorporating infected PBMC with and without anti-TNF-α neutralising antibodies. Consistent with our initial observation, anti-TNF-α neutralising antibodies suppressed the TNF-mediated increased Mtb growth, with a partial reduction at each concentration studied of two different neutralising antibodies (*Figure 5E* and *Figure 5—figure supplement 2*). Exogenous TNF-α modulated macrophage polarisation within microspheres, reducing CD80 expression at day 7 (*Figure 5—figure supplement 3*). Therefore, we next investigated whether anti-PD-1-induced Mtb growth could be reversed by blocking TNF-α activity. Anti-PD-1 antibody treatment of infected cells again augmented Mtb growth, and this increased growth could be reversed by the incorporation of anti-TNF-α antibodies into the

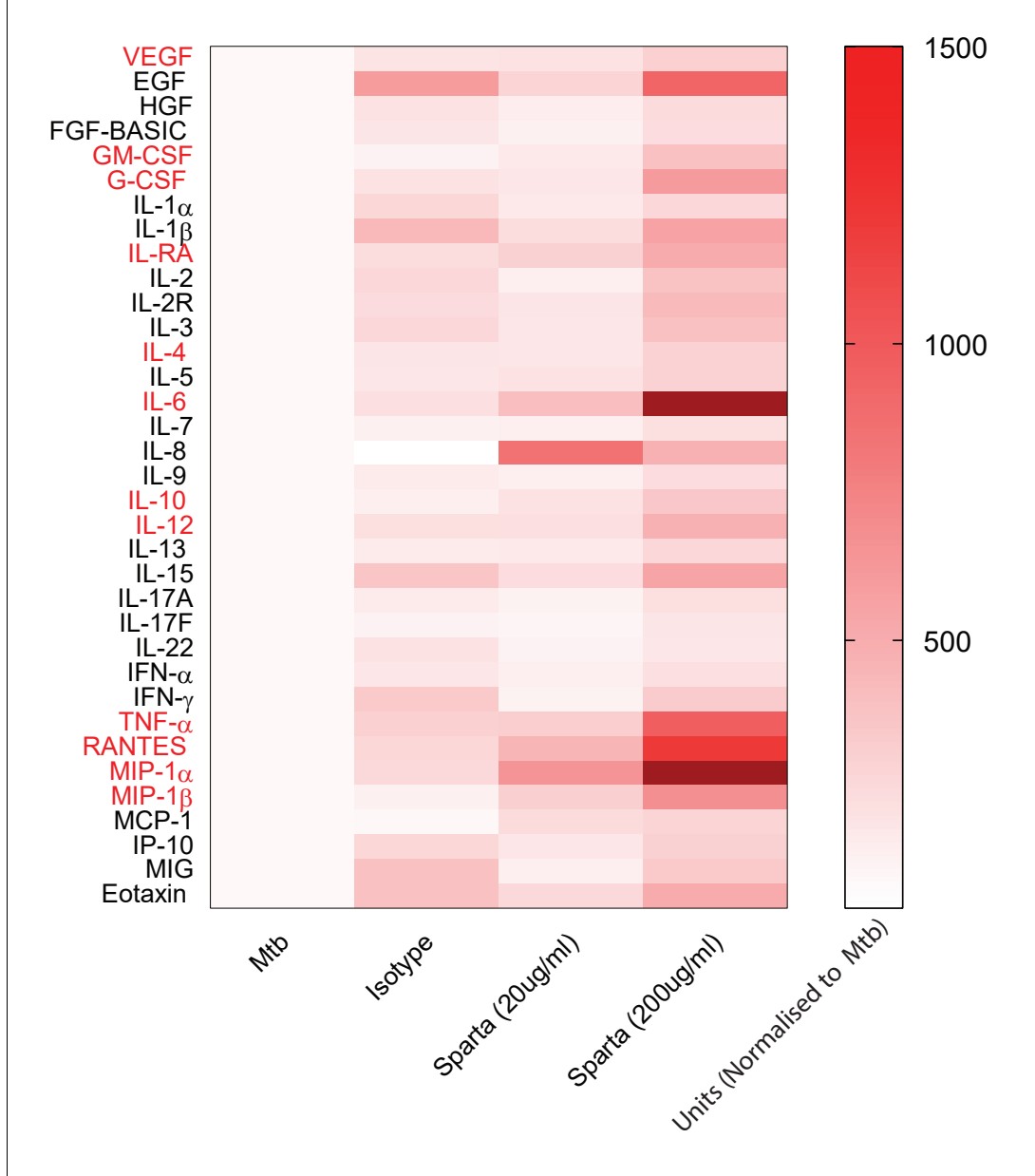

**Figure 4.** PD-1 inhibition increases secretion of multiple cytokines and growth factors. PD-1/PD-L1 signalling was inhibited by Spartalizumab, a humanized IgG4 anti-PD1 monoclonal antibody, in Mtb-infected microspheres at 20 and 200μ/ml in normoxia. Supernatants were collected at day 14 and accumulation of cytokines and growth factors was analysed by Luminex 35-multiplex assay. Concentrations were normalized to secretion by Mtb infected microspheres to demonstrate relative fold change, and individual concentrations are shown in *Figure 4—figure supplement 1*. The experiment was performed twice with three replicates. Red font: **p<0.001 for Spartalizumab versus isotype control.

The online version of this article includes the following figure supplement(s) for figure 4:

**Figure supplement 1.** Cytokine accumulation around microspheres after inhibition of PD-1/PD-L1 signalling with Spartalizumab, a humanized IgG4 anti-PD1 monoclonal antibody at 20 and 200 μg/ml in normoxia (N) and 1% hypoxia (H).

**Figure supplement 2.** PD-1 inhibition increases secretion of multiple cytokines and growth factors in 1% hypoxia.

**Figure supplement 3.** Spartalizumab induces TNF-α secretion in uninfected and infected microspheres, which is neutralised by anti-TNF-α.

microspheres (*Figure 5F*). This confirms that an excess of TNF-α is the primary driver of increased Mtb growth caused by PD-1 inhibition in the 3D model.

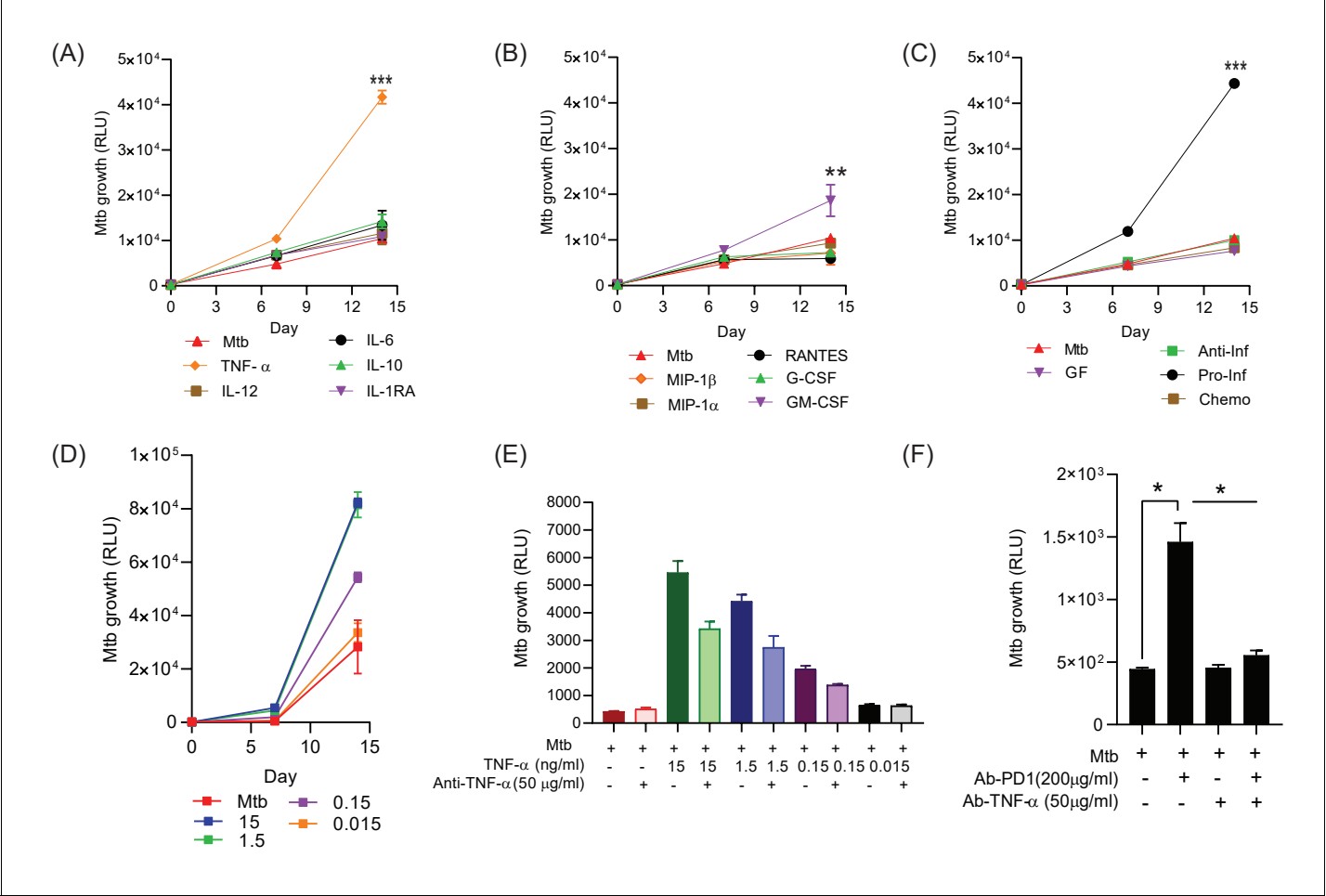

**Figure 5.** Addition of TNF-α increases Mtb growth in microspheres in normoxia. Recombinant human G-CSF, GM-CSF, IL-4, IL-6, IL-10, IL-12, TNF-α, IL-1RA, MIP-1α, MIP-1β or RANTES were added either individually (**A and B**) or in combination pools (**C**) to Mtb-infected microspheres at 'low' concentrations, defined as that measured in media around spheres after anti-PD-1 treatment. Recombinant human TNF-α increases growth of Mtb, whilst other pro-inflammatory cytokines did not (**A**). GM-CSF has a lesser growth-promoting effect (**B**). The only combination pool that increased Mtb growth was the pro-inflammatory cytokine pool, containing TNF-α (**C**). (**D**) TNF-α results in a dose-dependent increase in the Mtb growth over time. (**E**) Anti-TNF-α neutralising antibodies partially suppress the increased Mtb growth caused by TNF-α augmentation. Anti-TNF-α from Thermo Fisher Scientific. (**F**) Anti-PD1 antibody incorporation within microspheres increases of Mtb growth at day 7, and this effect is reversed by concurrent anti-TNF-α neutralising antibodies within microspheres. The constituent of the cytokine pools are: Growth factor pool (GF: GM-CSF and G-CSF), Anti-Inflammatory cytokine pool (Anti-Inf: IL-10 and IL-1RA), Pro-Inflammatory cytokine pool (Pro-Inf: TNF-α, IL-6 and IL-12) and Chemokine pool (Chemo: RANTES, MIP-1α, MIP-1β).

The online version of this article includes the following figure supplement(s) for figure 5:

**Figure supplement 1.** Individual Mtb growth curves at 'high' cytokine concentration, five times the concentration measured in media after anti-PD-1 treatment.

**Figure supplement 2.** Anti-TNF-α neutralizing antibodies supress the Mtb growth following TNF-α from a different source (Anti-TNF-α from Sigma-Aldrich, UK).

**Figure supplement 3.** TNF-α skews polarization of monocytes to macrophages with lower CD80 expression.

**Figure supplement 4.** Hierarchical gating strategy used to identify lymphocyte and monocytic populations from decapsulated microspheres containing human peripheral blood monocular cells.

## TNF-α is highly expressed in TB granulomas and sputum TNF-α negatively correlates with circulating PD-1 expression

Finally, to establish the in vivo relevance of our cell culture model findings, we performed immunohistochemical analysis of biopsies from patients with standard TB and anti-PD-1 associated TB. TNF-α was expressed within TB granulomas, with greater immunoreactivity than control lung tissue at the

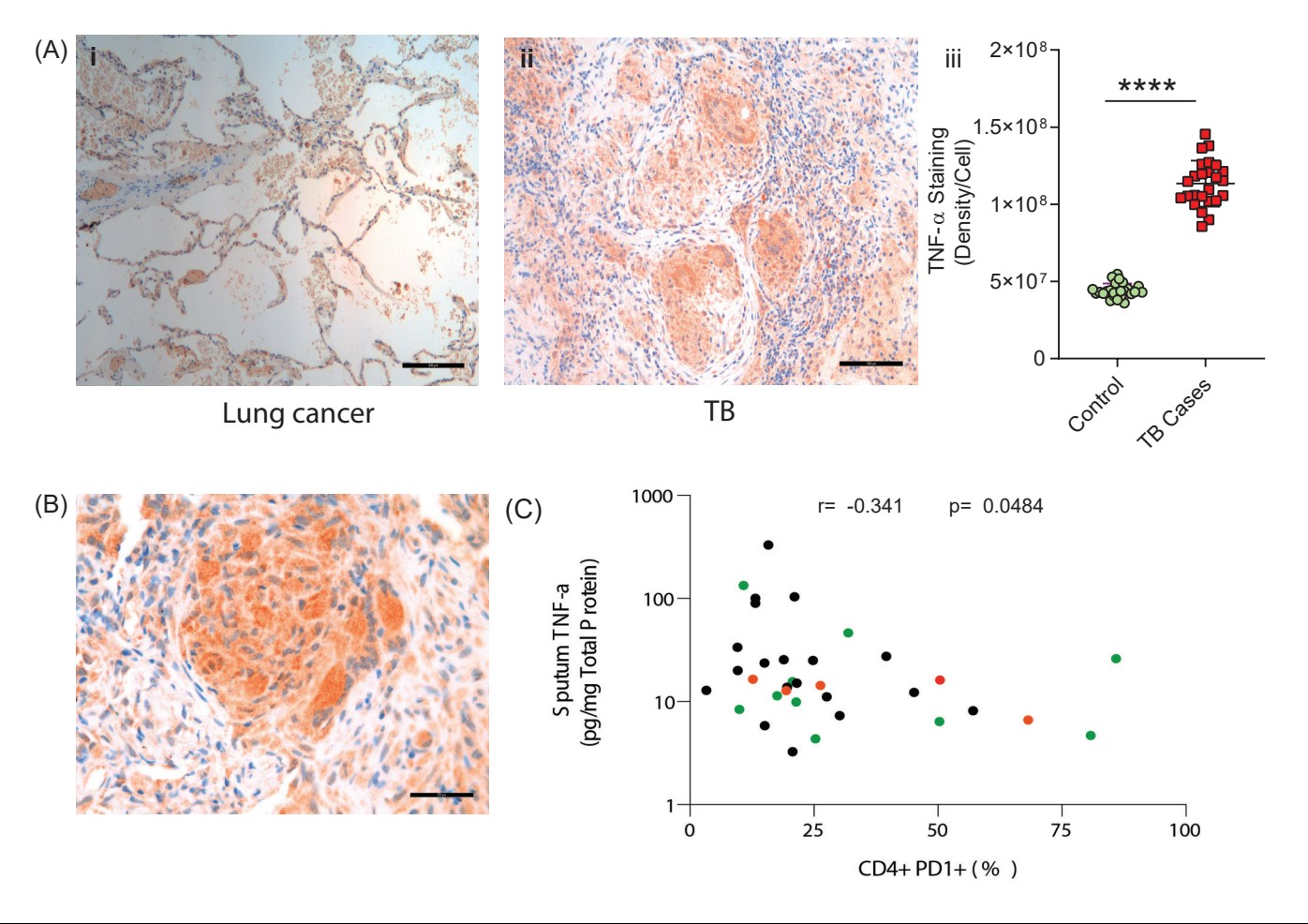

**Figure 6.** TNF-α is expressed in human TB granulomas and sputum TNF-α concentrations negatively correlate with circulating PD-1 expression. (**A**) TNF-α is expressed within human lung TB granulomas, with greater immunoreactivity than control lung tissue at the excision margin of lung cancer (i and ii). Quantification of TNF-α immunostaining (n = 5) in TB cases was significantly higher than controls (n = 5) (iii). (**B**) TNF-α immunostaining was extensive in the lung granuloma of a patient that developed TB whilst treated with pembrolizumab, a humanized anti-PD-1 antibody (n = 1). (**C**) CD4$^+$ T cell PD-1 expression on circulating PBMC negatively correlates with sputum TNF-α concentration in a separate cohort where paired sputum and PBMCs samples are available. Green dots, healthy controls; Black TB cases; Orange respiratory symptomatics. Analysis by Spearman's correlation analysis gave r-value of −0.341 with p=0.0484.

excision margin of lung cancer (*Figure 6A*). Consistent with our cell culture observations, TNF-α immunostaining was extensive in a biopsy from a patient that developed pulmonary TB whilst treated with pembrolizumab, an anti-PD-1 antibody (*Figure 6B*). Quantitative analysis of differences between standard TB and anti-PD-1 TB was not possible due to the unique nature of this clinical specimen, as TB diagnosis is usually made by bronchoalveolar lavage as opposed to percutaneous biopsy.

We then analysed the association between PD-1 expression by circulating CD4$^+$ and CD8$^+$ T cells and sputum TNF-α concentrations in a previously reported cohort (*Walker et al., 2017*). Although this involved comparing PD-1 expression by circulating T cells, remote from the site of disease, with total TNF-α concentration in the sputum, we hypothesised that a reverse association would support the conclusion that PD-1 limits TNF-α secretion in patients. Consistent with this, CD4$^+$ T cell PD-1

expression negatively correlated with sputum TNF-α, with a significant negative association despite the relatively low sample numbers, which would be expected to obscure an effect (*Figure 6C*).

## Discussion

Although Mtb kills more people than any other infection worldwide, an enduring enigma is that 90% of those exposed exert lifelong control of the pathogen. In seminal post-mortem studies when TB was highly prevalent in the United States, Opie showed that 30% of humans who died from other causes had viable Mtb present in the lung apices (*Opie and Aronson, 1927*). Therefore, in the majority of exposed individuals, a stable relationship forms between host immune cells and Mtb without disease developing. Disseminated TB disease often develops in immunocompromised individuals, such as advanced HIV infection, newborn infants or following anti-TNF-α therapy, and this observation informs the view that TB can arise as a disease of a deficient immune response to the pathogen (*O'Garra et al., 2013*).

However, the commonest form of human TB, and the one that leads to transmission, is apical pulmonary disease (*Elkington et al., 2011*). This occurs most frequently in young adults between the ages of 20–25 with the strongest recall response to Mtb antigens, as measured by the Mantoux test (*Comstock et al., 1974*). Therefore, these clinical observations demonstrate that infection results in a stable symbiosis between host and pathogen in the majority of individuals, and a pronounced immune response associates with the subsequent development of infectious pulmonary disease. The recently emerging clinical phenomenon of TB rapidly developing after initiating anti-PD-1 immunotherapy (*Fujita et al., 2016*; *Lee et al., 2016*; *Chu et al., 2017*; *Picchi et al., 2018*; *Jensen et al., 2018*; *Elkington et al., 2018*; *He et al., 2018*; *Takata et al., 2019*; *Barber et al., 2019*; *Tsai et al., 2019*; *van Eeden et al., 2019*) further reinforces that an excessive immune response in TB can be harmful.

PD-1 is expressed on T cells at the site of TB disease and PD-1 expression on circulating CD4+ T cells associates with bacterial load (*Day et al., 1995*). PD-1 expression is elevated in circulating CD4 T cells in TB (*Shen et al., 2016*) and has been proposed to limit an effective host immune response. Consequently PD-1 inhibition has been advanced as a therapeutic target to accelerate clearance of infection (*Zumla et al., 2016*; *Shen et al., 2016*; *Jurado et al., 2008*; *Singh et al., 2013*; *Suarez et al., 2019*; *Jiang et al., 2014*). However, from an evolutionary perspective, PD-1 is proposed to limit immunopathology in the face of chronic antigenic stimulation (*Sharpe and Pauken, 2018*). Therefore, it is equally plausible, and indeed perhaps more logical, that PD-1/PD-L1 pathway up-regulation in TB is a physiologically appropriate response to the persistent pathogen. We found that hypoxia further up-regulated the PD-1/PD-L1/2 axis, consistent with hypoxia increasing expression in cancer (*Noman et al., 2014*), and TB lesions are hypoxic both in model animals and human lesions (*Belton et al., 2016*; *Via et al., 2008*). Analysis of the effect of hypoxia is complicated that both host and pathogen physiology are altered, with hypoxia causing reduced Mtb growth (*Devasundaram et al., 2016*; *Ortega et al., 2014*; *Eoh and Rhee, 2013*) but also causing diverse host physiological changes. PD-1 may be particularly important in limiting excessive inflammation and pathology in conditions of low oxygen tension. TB reactivation following immune checkpoint blockade, and the extreme susceptibility of PD-1 deficient mice to Mtb infection (*Lázár-Molnár et al., 2010*; *Barber et al., 2011*), would support such a regulatory role, even though it runs counter to widely advanced disease paradigms.

Our work further highlights the double-edged sword of the host immune response in TB (*Ehlers, 1999*). TNF-α is clearly essential to an effective host immune response to TB, as disease frequently develops after treatment with anti-TNF-α antibodies (*Keane et al., 2001*), and TNF-α inhibits Mtb growth in zebrafish macrophages (*Clay et al., 2008*) and human alveolar macrophages (*Hirsch et al., 1994*). However, excessive TNF-α is also associated with poor outcomes in TB. In human cells, an excess of TNF-α can increase Mtb growth (*Byrd, 1997*; *Engele et al., 2002*), consistent with our observations. In patients with active TB, TNF-α expression by Mtb-specific T cells is increased (*Harari et al., 2011*; *Tebruegge et al., 2015*) and TNF-α associates with more severe radiological findings (*Casarini et al., 1999*; *Tsao et al., 2000*). In diverse model systems, excessive TNF-α has been shown to cause harmful inflammation (*Bekker et al., 2000*; *Taylor et al., 2005*; *Tsenova et al., 1999*; *Roca and Ramakrishnan, 2013*; *Tzelepis et al., 2018*), consistent with TNF-α exerting a bell-shaped effect on host immunity, with either deficit or excess detrimental. Similarly,

we report a deleterious effect of additional GM-CSF, in contrast to GM-CSF improving control of Mtb growth in murine macrophages (*Rothchild et al., 2014*). This suggests that cytokine responses are highly dose and context dependent, potentially with each demonstrating non-linear responses leading to a complex matrix of what may denote the optimal cytokine profile for Mtb control. Consistent with our findings, in the one patient that developed TB on anti-PD-1 treatment where longitudinal samples are available, there was a spike in PPD-specific TNF-$\alpha$ expressing cells prior to with the development of active TB (*Barber et al., 2019*), consistent with PD-1 acting as a regulator of TNF-$\alpha$ expression in TB. Taken together, these findings demonstrate a harmful effect of excessive TNF-$\alpha$ in TB.

We found that multiple cytokines and chemokines were increased after PD-1 inhibition, and therefore events in humans are likely to be more complex than TNF-$\alpha$ excess alone. The augmented inflammation may have multiple harmful effects, such as recruitment of excessive inflammatory cells and destruction of the extracellular matrix (*Cambier et al., 2017*; *Al Shammari et al., 2015*; *Mishra et al., 2017*), which favour Mtb growth. In the mouse model of TB, one key role of PD-1 is to limit IFN-$\gamma$ production (*Sakai et al., 2016*), and we have shown that excess IFN-$\gamma$ also accelerates Mtb growth in microspheres (*Tezera et al., 2017a*). We did not show an effect of TNF-$\alpha$ on cell survival, although TNF-$\alpha$ has been reported to increase macrophage necrosis in the zebrafish model, via a mitochondrial-lysosomal-endoplasmic reticulum circuit (*Roca and Ramakrishnan, 2013*; *Roca et al., 2019*). The effect of PD-1 inhibition on macrophage polarisation and survival warrants further investigation, as this is likely to be one determinant of outcome (*Lerner et al., 2017*; *Cronan et al., 2016*).

As in all in vitro systems, the bioelectrospray model has limitations (*Elkington et al., 2019*). For example, it does not permit the ingress of new inflammatory cells, and therefore can only be used to investigate the early events resulting from anti-PD-1 treatment, and not the recruitment of inflammatory cells by increased chemokine secretion. In addition, although it permits longer analysis of the host-pathogen interaction than other human primary cell culture systems, the 3 week standard experiment remains shorter than human infection, and so it models early events. We have not yet characterised which cells produce TNF-$\alpha$, nor the wider phenotypic changes that result from inhibition of the PD-1 axis. The optimal approach would be to integrate single cell RNAseq to the analysis pipeline, so the phenotype of different cells can be comprehensively analysed. As microspheres can be readily dissolved by incubation in EDTA to release the cells, the system is suited for this technical development.

The common side effects of anti-PD-1 treatment in patients are termed immune related Adverse Events (irAEs), which are autoimmune in nature (*Postow et al., 2018*). We have proposed that an autoimmune process may exacerbate pathology in TB (*Elkington et al., 2016*), and others have suggested that a loss of tolerance underlies progression to active TB (*Divangahi et al., 2018*; *Olive and Sassetti, 2018*). The common theme is that an excessive response to antigens, whether host or pathogen-derived, can drive disease in TB and our findings further support this conclusion. PD-1 may act to fine tune the balance between pro- and anti-inflammatory responses necessary to control infection without causing pathology. Notably, immune related adverse events to immune checkpoint inhibitors may be treated with anti-TNF-$\alpha$ antibodies (*Postow et al., 2018*), suggesting TNF-$\alpha$ may be the primary driver of both autoimmunity and TB pathology after PD-1 treatment.

Our model provides the mechanistic insights into a clinical phenomenon with significant implications for future TB treatment and vaccine approaches. Simply driving a stronger immune response to Mtb seems unlikely to be beneficial, as clinical and epidemiological data suggest it may be harmful. For example, host-directed therapies may be designed to accelerate bacterial clearance whilst concurrently reducing immunopathological effects by appropriate skewing of macrophage phenotype. A more nuanced view considering the different immunological phases of TB is essential, differentiating events at the point of initial exposure from the late events at the apex of the lung, where excessive inflammation leads to immunopathology and transmission (*Elkington and Friedland, 2015*). Defining the distinction between a protective and pathological immune response in human TB remains a key unanswered question, essential to inform new interventions to control the TB pandemic.

# Materials and methods

**Key resources table**

| Reagent type (species) or resource | Designation | Source or reference | Identifiers | Additional information |
|---|---|---|---|---|
| Strain, strain background (*Mycobacterium tuberculosis*) | H37Rv | (From Ref: 20) | | Used at multiple of infection of 0.1 |
| Strain, strain background (*Mycobacterium tuberculosis*) | H37Rv pMV306hsp+ LuxAB+G13+CDE | (From Ref: 20) | | Used at multiple of infection of 0.1 |
| Strain, strain background (*Mycobacterium tuberculosis*) | H37Rv pMV306hsp encoding the wild-type FFluc | (From Ref: 20) | | Used at multiple of infection of 0.1 |
| Biological sample (Antibodies) | | | | |
| Antibody | anti-CD45-V500 Horizon (Clone no. HI30) | BD Biosciences | Cat.No.563792 | Monoclonal mouse antibody |
| Antibody | anti-CD3 Brilliant Violet 785 (Clone no. OKT3) | Biolegend | Cat.No.317330 | Monoclonal mouse antibody |
| Antibody | anti-CD3-PE (Clone no. HIT3a) | Biolegend | Cat.No.300308 | Monoclonal mouse antibody |
| Antibody | anti-CD4 Brilliant Ultra Violet 496 (Clone no. SK3) | BD Bioscience | Cat.No.612937 | Monoclonal mouse antibody |
| Antibody | anti-CD4-PerCP (Clone no. OKT4) | Biolegend | Cat.No.317432 | Monoclonal mouse antibody |
| Antibody | anti-CD8 Brilliant Violet 605 (Clone no. RPA-T8) | Biolegend | Cat.No.301040 | Monoclonal mouse antibody |
| Antibody | anti-CD8-APC (Clone no. SK1) | Biolegend | Cat.No.344722 | Monoclonal mouse antibody |
| Antibody | anti-CD103-APC (Clone no. Ber-ACT8) | BD Biosciences | Cat.No.563883 | Monoclonal mouse antibody |
| Antibody | anti-CD69 Brilliant Ultra Violet 395 (Clone no. FN50) | Biolegend | Cat.No.310902 | Monoclonal mouse antibody |
| Antibody | anti-HLA-DR-PerCP (Clone no. L243) | Biolegend | Cat.No.307628 | Monoclonal mouse antibody |
| Antibody | anti-CD279-BB515 (Clone no. EH12.1) | BD Biosciences | Cat.No.564494 | Monoclonal mouse antibody |
| Antibody | anti- CD-274-BB515 (Clone no. MIH1) | BD Biosciences | Cat.No.564554 | Monoclonal mouse antibody |
| Antibody | anti-PD-1 Brilliant Violet 421 (Clone no.EH12.1) | BD Biosciences | Cat.No.562516 | Monoclonal mouse antibody |
| Antibody | anti-CD11b-APC (Clone no. ICRF44) | Biolegend | Cat.No.301310 | Monoclonal mouse antibody |
| Antibody | anti-CD45-APC/Cy7 (Clone no. 2D1) | Biolegend | Cat.No.368516 | Monoclonal mouse antibody |
| Antibody | anti-CD14-AP/APC (Clone no. HCD14) | Biolegend | Cat.No.325608 | Monoclonal mouse antibody |
| Antibody | anti-True-Stain Monocyte Blocker | Biolegend | Cat.No.426102 | Monoclonal mouse antibody |
| Antibody | anti-CD4 (Clone no. M7310) | DAKO | Cat.No.M7310 | Monoclonal mouse antibody |

*Continued on next page*

*Continued*

| Reagent type (species) or resource | Designation | Source or reference | Identifiers | Additional information |
|---|---|---|---|---|
| Antibody | anti-CD8 (Clone no. M7103) | DAKO | Cat.No. M7103 | Monoclonal mouse antibody |
| Antibody | anti-PD1 (Clone no. ab5287) | Abcam | Cat.No.ab52587 | Monoclonal mouse antibody |
| Antibody | anti-TNF-α (Clone no. ab1793) | Abcam | Cat.No.ab1793 | Monoclonal mouse antibody |
| Antibody | Spartalizumab | Selleckchem | Cat.No.A2017 | 20 µg/ml and 200 µg/ml, monoclonal, mouse IgG4 |
| Antibody | IgG4 | Sino Biologicals | Cat.No.13505-HNAH | 20 µg/ml and 200 µg/ml, monoclonal, mouse IgG4 |
| Antibody | Mouse IgG1 kappa Isotype Control (P3.6.2.8.1), | Thermo Fisher Scientific | Cat.No.16-4714-82 | 50 µg/ml, mouse monoclonal IgG2A |
| Antibody | Mouse IgG1 Negative Control, clone Ci4 | Merck Life Sciences | Cat.No.MABC002 | 51 µg/ml, mouse monoclonal IgG2A |
| Antibody | anti-TNF-α | Thermo Fisher Scientific | Cat.No.16-7348-81 | 52 µg/ml, mouse monoclonal IgG2A |
| Sequence-based reagent (Applied Biosystems TaqMan Gene Expression primers) | GAPDH | Thermo Fisher Scientific | #Hs02758991_g1 | |
| | β2-Microbulin | Thermo Fisher Scientific | #Hs00608023_m1 | |
| | FNTA | Thermo Fisher Scientific | #Hs00357739_m1 | |
| | PDCD1 | Thermo Fisher Scientific | #Hs01550088_m1 | |
| | CD274 | Thermo Fisher Scientific | #Hs00204257_m1 | |
| | PDCD1LG2 | Thermo Fisher Scientific | #Hs00228839_m1 | |
| | HIF-1α | Thermo Fisher Scientific | #Hs00153153_m1 | |
| Commercial assay or kit | CytoTox-Glo Cytotoxicity Assay | Promega | G9291 | Commercial assay or kit |
| | Lactate Dehydrogenase Activity Assay Kit | Merck | 11 644 793 001 | Commercial assay or kit |
| | Cytokine and Chemokine 35-Plex Human ProcartaPlex Panel | Thermo Fisher Scientific | LHC6005M | Commercial assay or kit |
| Chemical compound, drug | PD-1/PD-L1 Inhibitor 1 | Cambridge Biosciences, UK | #1675201-83-8 | Chemical compound |
| | Recombinant Human G-CSF | ImmunoTools | Cat.No.11343115 | 1 ng/ml and 5 ng/ml |
| | Recombinant Human GM-CSF | ImmunoTools | Cat.No.11343125 | 0.25 ng/ml and 1.25 ng/ml |
| | Recombinant Human IL-1RA/IL1 F3 | ImmunoTools | Cat.No.11344876 | 1.25 ng/ml and 6.25 ng/ml |
| | Recombinant Human IL-10 | ImmunoTools | Cat.No.11340105 | 0.2 ng/ml and 1 ng/ml |
| | Recombinant Human IL-6 | ImmunoTools | Cat.No.11340066 | 10.0 ng/ml and 50.0 ng/ml |
| | Recombinant Human IL-12 | ImmunoTools | Cat.No.11349125 | 0.5 ng/ml and 2.5 ng/ml |

*Continued on next page*

*Continued*

| Reagent type (species) or resource | Designation | Source or reference | Identifiers | Additional information |
|---|---|---|---|---|
| | Recombinant Human TNF-α | ImmunoTools | Cat.No.11343017 | 0.3 ng/ml, 1.5 ng/ml, 7.5 ng/ml and 15 ng/ml |
| | Recombinant Human IL-15 | ImmunoTools | Cat.No.11340155 | 0.5 ng/ml, 5 ng/ml and 50 ng/ml |
| | Recombinant Human IL-17A | ImmunoTools | Cat.No.11340176 | 1 ng/ml, 10 ng/ml and 100 ng/ml |
| | Recombinant Human IL-17F | ImmunoTools | Cat.No.11349176 | 1 ng/ml, 10 ng/ml and 100 ng/ml |
| | Recombinant Human RANTES | ImmunoTools | Cat.No.11343196 | 0.3 ng/ml and 1.5 ng/ml |
| | Recombinant Human MIP-1α | ImmunoTools | Cat.No.11343206 | 1.5 ng/ml and 7.5 ng/ml |
| | Recombinant Human MIP-1β | ImmunoTools | Cat.No.11343223 | 1.0 ng/ml and 5 ng/ml |
| | Recombinant Human MCP | ImmunoTools | Cat.No.11343386 | 1 ng/ml, 10 ng/ml and 100 ng/ml |
| Software, algorithm | FlowJo | BD Bioscences | version 10.6.1 | Software |
| | BD FACSDiva Software | BD Biosciences | | Software |
| | Graphpad Prism | GraphPad Software LLC | v7.05 | Software |

## *M. tuberculosis* culture

*M. tuberculosis* H37Rv (Mtb) was cultured in Middlebrook 7H9 medium (supplemented with 10% ADC, 0.2% glycerol and 0.02% Tween 80) (BD Biosciences, Oxford). Bioluminescent Mtb containing luxABCDE (Mtb lux+) and Mtb expressing ffLuc (Mtb ffLuc+) were cultured with kanamycin 25 µg/ml. Luminescence was measured with either GloMax 20/20 single tube luminometer (Promega,UK) or GloMax Discover microplate reader (Promega,UK). Cultures at $1 \times 10^8$ CFU/ml Mtb (OD = 0.6) were used for all experiments at multiplicity of infection (MOI) of 0.1. Live Mtb was used in all experiments apart from the time lapse microscopy, which used UV killed TB. Mtb colony counting was performed by serial dilution on Middlebrook 7H11 Agar. Bioluminescence from the Mtb ffLuc+ was induced using D-luciferin (ThermoFisher, UK) at a concentration of 750 µM in Hank's balanced salt solution (HBSS).

## PBMC cell isolation from human blood

For the 3D microsphere experiments, PBMC were separated from single donor leukocyte cones (National Health Service Blood and Transfusion, Southampton, UK) by density gradient centrifugation over Ficoll-Paque (GE Healthcare Life Sciences). Ethical approval for these studies was provided by the National Research Ethics Service Committee South Central - Southampton A, ref 13/SC/0043.

Study participants for the correlation analysis of sputum-TNF-α with PD-1 expression on CD4+ and CD8+ T cells are the cross-sectional study individuals in a previously reported cohort (*Walker et al., 2017*; *Walker et al., 2019*). Participants were recruited from an outpatient clinic in Khayelitsha, South Africa and were either healthy volunteers, non-TB respiratory symptomatics or recently diagnosed TB patients. Flow cytometric analysis was performed on cryopreserved PBMC isolated from whole blood as previously reported (*Walker et al., 2019*). The study was approved by the University of Cape Town Human Research Ethics Committee (REF 516/2011) and conducted in accordance with the Declaration of Helsinki.

## Analysis of PD-1 expression in blood and lung of TB patients

Lung tissue and matched PBMC were obtained from the AHRI Lung study cohort approved by the Biomedical Research Ethics Committee (BREC) of the University of Kwa-Zulu Natal, BREC reference:

BE019/13. All participants underwent surgical resection to treat TB related lung complications, including haemoptysis, bronchiectasis, persistent cavitatory disease, shrunken or collapsed lung or drug-resistant infection, at the King Dinuzulu Hospital in Durban, KwaZulu- Natal and Inkosi Albert Luthuli Central Hospital (IALCH) in Durban, KwaZulu-Natal. PBMC were isolated from whole blood using standard Ficoll-Histopaque (Sigma) density gradient centrifugation by standard protocol.

Lung tissues was cut into approximately 1 mm$^3$ pieces, washed several times with cold HBSS (Lonza) and re-suspended in 8mls of pre-warmed digestion media R10 (RPMI supplemented with 10% FCS, 2 mM L-glutamate, 100 U/ml Penstrep), containing 0.5 mg/ml collagenase D (Roche) and 40 U/ml DNaseI (Roche), and transferred to GentleMACS C-tubes (Miltenyi) for mechanical digestion according to the manufacturer's instructions. The suspension was incubated for 30 min at 37°C, followed by an additional mechanical digestion step and another 30 min incubation step at 37°C. The final suspension was passed through a 70 µm cell strainer and washed twice in HBSS. PBMC and lung cells were phenotyped by surface staining with a near-infrared live/dead cell viability cell staining kit (Invitrogen) and a cocktail of fluorochrome conjugated antibodies: αCD45-V500 Horizon clone HI30 (BD Biosciences), αCD3 Brilliant Violet 785 clone OKT3 (Biolegend), αCD4 Brilliant Ultra Violet 496 clone SK3 (BD Bioscience), αCD8 Brilliant Violet 605 clone RPA-T8 (Biolegend), αCD103-APC clone Ber-ACT8 (BD Biosciences), αCD69 Brilliant Ultra Violet 395 clone FN50 (Brilliant Horizon), αPD-1 Brilliant Violet 421 clone EH12.1 (BD Biosciences). Cells were stained with 25 uL of antibody cocktail in the dark for 20 min at room temperature followed by washing with PBS, then fixed in 2% PFA. Data was acquired using BD Aria Fusion cytometer and analyzed using FlowJo Software v.9.9 (Treestar Inc, Ashland, OR).

## Immunohistochemistry of paraffin-fixed tissue

Immunohistochemical analysis was performed on paraffin-embedded lung tissue from patients with pulmonary TB and lung cancer that were mounted at 4 µm thin onto APS coated glass slides and dried, using the following antibodies: anti-CD4 (Clone no. M7310) (DAKO), anti-CD8 (Clone no. M7103) (DAKO), anti-PD1 (Clone no. ab5287) (Abcam) and anti-TNF-α (Clone no. ab1793) (Abcam). Staining was done at optimised concentrations usingrecommended buffers for each antibody.

## Microencapsulation of cells

Microspheres were generated with an electrostatic generator (Nisco, Zurich, Switzerland) as described previously (*Tezera et al., 2017b*). Briefly, PBMC were infected overnight with Mtb in a 250 cm$^2$ flask, cells were detached, pelleted and mixed with 1.5% sterile alginate (Pronova UP MVG alginate, Nova Matrix, Norway) and 1 mg/mL collagen (Advanced BioMatrix, USA) at a final concentration of $5 \times 10^6$ cells/ml. The cell-alginate suspension was injected into the bead generator where microspheres were formed in an ionotropic gelling bath of 100 mM CaCl$_2$ in HBSS. After washing twice with HBSS with Ca$^{2+}$/Mg$^{2+}$, microspheres were transferred in RPMI 1640 medium containing 10% human AB serum and incubated at 37°C. Microspheres were either dispensed into eppendorfs, which were then randomly allocated to different environmental conditions, or plated into a 96-well plate with conditions in triplicate according to a pre-determined template. For experiments in hypoxia, microspheres were incubated in 1% oxygen in Galaxy 48 R CO$_2$ incubator (Eppendorf, UK) until analysis. Supernatants were collected at defined time points. Time points described are days post infection.

## Live cell imaging

Uninfected or UV killed Mtb infected cells suspended in alginate-collagen matrix were plated in an eight well µ-Slide (ibidi GmbH, Germany). HBSS containing Ca$^{2+}$ was used for cross-linking of alginate in the extracellular matrix for 15 min and then replaced with RPMI medium with 10% human AB serum. Samples were imaged using an Olympus IX81 time-lapse microscope with temperature of 37°C and CO$_2$ concentration 5%. Z-stacks 200 µm in height were captured at one position in each sample every 30 min for 48 hr. Images were exported as tif files and opened in ImageJ.

## PD-1/PD-L1 Inhibition

Small chemical inhibition of PD-1/PD-L1 signalling was by PD-1/PD-L1 Inhibitor 1 (CAS Registry #1675201-83-8, Cambridge Biosciences, UK), a compound that competitively blocks the interaction

of PD-1 with it ligand protein PD-L1 (*Guzik et al., 2017*). Inhibitor one was prepared in DMSO (Sigma-Aldrich,UK) at a concentration of 6.3 mM and dissolved to a concentration of 1 nM, 10 nM, 100 nM and 1 µM in complete media and added to media around microspheres on the day after encapsulation. Luminescence was monitored on specific days. For the microspheres incubated in hypoxic conditions, measurement of luminescence was repeated for every 30 min after the addition of luciferin until the reading plateaued, which was usually 2 hr.

In antibody inhibitory experiments, spartalizumab (Selleckchem, Germany), a humanised IgG4 anti-PD1 monoclonal antibody, was used to inhibit of PD-1/PD-L1 signalling in microspheres. Briefly, cells were infected with Mtb overnight, pelleted and then a suspension of anti-PD1 antibodies (20 and 200 µg/ml) were added and pre-incubated for 1 hr. Cells were then encapsulated within microspheres and kept for 14 days in either normoxia or 1% oxygen at 37°C and 5% $CO_2$. Mtb growth was measured using luminescence and supernatants were taken for either cytokine or LDH measurement. An IgG4 human antibody was used at the same concentration as a control.

## Cell toxicity assays

Lactate dehydrogenase (LDH) release in the supernatants collected at different time points was analysed by a colorimetric activity assay as per manufacturer's instructions (Roche, Burgess Hill, United Kingdom). As a second assay, CytoTox-Glo Cytotoxicity Assay (Promega) was used, which measures dead-cell protease activity released from cells without membrane integrity using a luminogenic peptide substrate, the AAF-Glo Substrate. Luminescence from 96-well plates was analysed by GloMax Discover (Promega). The LDH assay was suited for later time points, as this could be performed on microspheres in eppendorfs, while the CytoTox glow required analysis in 96 well plates and so was best suited to analysis in the first week.

## Gene expression analysis

All the reagents were sourced from ThermoFisher Scientific (Paisley, UK). In brief, microspheres were decapsulated with 5 mM EDTA, cells were pelleted and immediately lysed using TRIzol Reagent. RNA was transcribed using High Capacity cDNA Reverse Transcription kit. TaqMan Universal master mix and primers specific for genes were GAPDH (#Hs02758991_g1), β2-Microbulin (#Hs00608023_m1), FNTA (#Hs00357739_m1), PDCD1 (#Hs01550088_m1), CD274 (#Hs00204257_m1), PDCD1LG2 (#Hs00228839_m1) and HIF-1α (Hs00153153_m1) were used for qPCR according to the manufacturer's instructions and the comparative threshold (CT) method was employed to analyse all qPCR data.

## Immunophenotyping of cells from microspheres

Microspheres were decapsulated with 5 mM EDTA in HBSS with no $Ca^{2+}/Mg^{2+}$ at day seven after encapsulation and 2 million cells prepared for staining in RPMI with 5% foetal bovine serum. To measure the expression of PD-1 in $CD4^+$ and $CD8^+$ T cells and PD-L1 expression in $CD14^+CD11b^+$ cells, the following antibody panel was used: CD3-PE (clone HIT3a, Biolegend), HLA-DR-PerCP (clone L243, Biolegend), CD4-PerCP (clone OKT4, Biolegend), CD8-APC (clone SK1, Biolegend), CD11b-APC (clone ICRF44, Biolegend), CD45-APC/Cy7 (clone 2D1, Biolegend), CD14-AP/APC (clone HCD14, Biolegend), CD279-BB515 (Clone EH12.1, BD), CD-274-BB515 (Clone MIH1, BD) and True-Stain Monocyte Blocker (Biolegend, USA). Gates were defined using fluorescence minus one control after exclusion of the dead cells using Live/Dead fixable stain (ThermoFisher, UK). Gating strategy is provided in *Figure 5—figure supplement 4*. Cells were acquired after fixing them in 2% paraformaldehyde in HBSS for 1 hr using FACSAria (Becton Dickinson, UK) and analysed by FACSDiva software (Becton Dickinson) and Flow Jo version 10 (Treestar).

## Cytokine supplementation

Microspheres were incubated in RPMI 1640 with 10% AB serum in an opaque 12-well tissue culture plate with G-CSF, GM-CSF, IL-1RA, IL-6, IL-10, IL-12, IL-15, IL-17A, IL-17F, TNF-α, RANTES, MIP-1α, MIP-1β and MCP at two concentrations determined by the cellular experiments, at 37°C and 5% $CO_2$. All the cytokines were purchased from ImmunoTools (Germany), suspended in RPMI with 0.1% human serum and kept at −80°C until use. Bacterial growth was monitored with luminescence using GloMax Discover microplate reader (Promega,UK).

## Luminex analysis

Samples were sterilised by filtration through a 0.22 µM Durapore membrane (MerkMillipore). Concentrations of cytokines (ThermoFisher, UK) were determined using a Bioplex 200 platform (Bio-Rad, UK) according to the manufacturer's protocol and quantified per milligram of total protein measured by Bradford assay (Biorad).

## Statistical analysis

All experiments were performed on a minimum of 2 occasions from separate donors as biological replicates and on each occasion with a minimum of 3 technical replicates. Some donor-to-donor variation occurred in terms of absolute RLU, as expected in the analysis of primary human cells, but the direction of effects were always consistent. Data presented are from a representative donor and include the mean and SEM. Analysis was performed in Graphpad Prism v7.05. Students t-test was used to compare pairs and ANOVA with Tukey's correction for multiple comparisons for groups of 3 or more groups where it was appropriate. For the flow cytometric analysis of clinical samples, data were analysed using Mann-Whitney test for comparing pairs and Kruskal-Wallis test with Dunn's multiple comparisons test for three or more group.

# Acknowledgements

This work was supported by the UK Medical Research Council MR/N006631/1 and MR/P023754/1 (PE), an Innovation Grant 2017 from Wessex Medical Research and a Postdoctoral Career Track Award from University of Southampton (LT). AL was supported by BMGF (OPP1137006) and the Wellcome Trust (210662/Z/18/Z). SM was supported by Cancer Research UK (23562) and UK Medical Research Council (MR/S024220/1). RJW and KAW receive support from the Francis Crick Institute, which is funded Wellcome Trust (FC0010218), UK research and Innovation (FC0010218), and Cancer Research UK (FC0010218). RJW also receives support from Wellcome Trust (203135, 104803) and NIH, USA (U19AI111276). NFW was supported by an NIHR Academic Clinical Lectureship. We also thank the Department of Infection Biology at London School of Hygiene and Tropical Medicine for access to facilities. We thank Jennifer Russell, Regina Teo and Monette Lopez, University of Southampton, for excellent technical assistance. We thank Siouxsie Wiles for providing the Lux-expressing Mtb.

# Additional information

## Funding

| Funder | Grant reference number | Author |
| --- | --- | --- |
| Medical Research Council | MR/P023754/1 | Paul T Elkington |
| Medical Research Council | MR/N006631/1 | Paul T Elkington |
| Wessex Medical Research | Innovation Grant 2017 | Liku B Tezera |
| Wellcome Trust | 210662/Z/18/Z | Alasdair Leslie |
| Bill and Melinda Gates Foundation | OPP1137006 | Alasdair Leslie |
| Cancer Research UK | 23562 | Salah Mansour |
| Medical Research Council | MR/S024220/1 | Salah Mansour |
| Wellcome Trust | FC0010218 | Robert J Wilkinson Katalin Andrea Wilkinson |
| UK Research and Innovation | FC0010218 | Robert J Wilkinson Katalin Andrea Wilkinson |
| Cancer Research UK | FC0010218 | Robert J Wilkinson Katalin Andrea Wilkinson |
| Wellcome Trust | 203135 | Robert J Wilkinson |
| Wellcome Trust | 104803 | Robert J Wilkinson |

| National Institutes of Health | U19AI111276 | Robert J Wilkinson |
| National Institute for Health Research | Academic Clinical Lectureship | Naomi F Walker |

The funders had no role in study design, data collection and interpretation, or the decision to submit the work for publication.

## Author contributions

Liku B Tezera, Conceptualization, Data curation, Formal analysis, Funding acquisition, Investigation, Methodology, Project administration; Magdalena K Bielecka, Formal analysis, Investigation; Paul Ogongo, Naomi F Walker, Diana J Garay-Baquero, Sanjay Jogai, Suwan N Jayasinghe, Salah Mansour, Gareth J Thomas, Christian H Ottensmeier, Data curation, Formal analysis; Matthew Ellis, Formal analysis; Kristian Thomas, Michaela T Reichmann, Mohamed Ahmed, Data curation; David A Johnston, Data curation, Methodology; Katalin Andrea Wilkinson, Data curation, Investigation; Robert J Wilkinson, Data curation, Formal analysis, Funding acquisition; Alasdair Leslie, Conceptualization, Data curation, Formal analysis, Supervision, Funding acquisition; Paul T Elkington, Conceptualization, Data curation, Formal analysis, Funding acquisition, Investigation, Project administration

## Author ORCIDs

Liku B Tezera (iD) https://orcid.org/0000-0002-7898-6709
Paul Ogongo (iD) http://orcid.org/0000-0002-0093-5768
Diana J Garay-Baquero (iD) http://orcid.org/0000-0002-9450-8504
Michaela T Reichmann (iD) http://orcid.org/0000-0002-6714-8400
Katalin Andrea Wilkinson (iD) http://orcid.org/0000-0002-9796-2040
Salah Mansour (iD) http://orcid.org/0000-0002-5982-734X
Gareth J Thomas (iD) http://orcid.org/0000-0003-3832-7335
Christian H Ottensmeier (iD) http://orcid.org/0000-0003-3619-1657
Paul T Elkington (iD) https://orcid.org/0000-0003-0390-0613

## Ethics

Human subjects: All ethical approvals were in place from the appropriate regulatory organisations in both the UK and South Africa, as cited in the methods.

## Decision letter and Author response

Decision letter https://doi.org/10.7554/eLife.52668.sa1
Author response https://doi.org/10.7554/eLife.52668.sa2

# Additional files

## Supplementary files

- Source data 1. Primary data for figures provided in manuscript.
- Supplementary file 1. PD-1 expressing cells for each subset expressed as percentage of live CD45 + cells, with range in parentheses.
- Transparent reporting form

## Data availability

All data generated or analysed during this study are included in the manuscript and supporting files. Source data files have been provided for for all figures as a data resource file.

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
