## [Decision Letter]

**Acceptance summary:**

This work uses an innovative microsphere 3-D model to dissect host-pathogen interactions during human infection with *Mycobacterium tuberculosis* by investigating the impact of PD-1 inhibition on infection outcome. With the microsphere model, the authors show that inhibition of PD-1 is detrimental to host control of Mtb due to excessive TNF production, thereby contributing to tuberculosis susceptibility from PD1 blockers.

**Decision letter after peer review:**

Thank you for submitting your article "Anti-PD-1 immunotherapy leads to tuberculosis reactivation via dysregulation of TNF-α" for consideration by *eLife*. Your article has been reviewed by three peer reviewers, and the evaluation has been overseen by a Reviewing Editor and Tadatsugu Taniguchi as the Senior Editor. The following individual involved in review of your submission has agreed to reveal their identity: Alissa Rothchild (Reviewer #2).

The reviewers have discussed the reviews with one another and the Reviewing Editor has drafted this decision to help you prepare a revised submission.

Summary:

Tezera et al. extend previous findings using a 3D biospray cell culture model to dissect host-pathogen interactions during human Mtb infection by investigating the impact of PD-1 blockade on infection. The authors demonstrate that PD-1 is expressed on lung resident T cells from TB patients and that PD-1 and PD-L1 expression increases following infection of PBMCs in the 3D model. They show that inhibition of PD-1 is detrimental to host control of Mtb due to excessive TNF production. This is an interesting paper, but as the authors themselves point out, it is already known that PD-1 blockade leads to an increase in cytokine production and loss of Mtb control. The dogma thus far, is that excessive Th1 responses are the mechanism of TB susceptibility in PD-1 blockers cases. This paper fights this dogma; and says that the pro-inflammatory cytokines drive Mtb replication, and this excessive cytokine response is what is at the root of TB susceptibility from PD1 blockers. The authors have designed a number of experiments with their innovative three-dimensional model, called the microsphere.

The reviewers were enthusiastic about the novel application of microsphere to this contemporary issue, but given that other studies have demonstrated a clear roles for TNF during Mtb infection, they raised concerns about lack of mechanistic detail as to how PD-1 inhibition increases TNF production in this 3D model and what impact excessive TNF is having on immune cell interactions in the microsphere system. In addition, some results (see below) need further data and statistics to support the conclusions.

Essential revisions:

1) Figure 1: PD-1 is expressed in granulomas but not in areas of immunopathology. This is an important point being made, with human samples, but the first statistical testing is PD1 levels in tissue cells versus blood cells. Perhaps a) cells become PD1 positive in tissue, b) PD1 cells hone to the tissue, c) other? Though p-values are reported, it wasn't clear what the hypothesis was. It was much clearer in the tissue sections when the authors set out to determine whether the PD1+ were and were not, in the tissue. Maybe the statistical testing should be reserved for the data that directly address the experimental hypotheses? In Figure 1A, there are ~ 35 dots in the lung plot with a median frequency of PD1+ cells of about 15%. Does this mean that the authors studied 35 independent samples from 35 patients? Please describe the N for the pathology samples in the Results (perhaps this was in a supplementary file?). It seems that the biologic difference between blood and lung is very thin and the statistical difference is driven by the large number of dots. It would be important to clarify the number of independent lung samples studied for this figure. Finally, in the text, the authors conclude by saying: "PD-1 is expressed by lung resident T-cells and may be required to prevent destructive lung disease." I believe this is based on an inverse correlation. It might be preferable to conclude the section with what is observed, then introduce the hypothesis in the next section. There are other critical questions that must be addressed in Figure 1. How does the frequency of PD-1 expressing T cells in TB patients compare to healthy controls? It would be helpful in Figure 1A to compare the frequencies (at least in the blood) to those in healthy controls to understand whether the increase in PD-1 is lung specific or systemic during Mtb infection. Second, in Figure 1B, PD-1 expression appears to be highest on the CD103+ CD69+ population for CD8^+^ T cells, but CD103 does not seem to mark the CD4^+^ T population with the highest PD-1 expression. Is CD103 known to mark tissue residency for CD4^+^ T cells like it does for CD8^+^ T cells? If not, it might be better to leave this marker out of the analysis in Figure 1B-I for clarity. It would also be useful to include the overall frequencies of each of the four CD103/CD69 populations for both CD4^+^ and CD8^+^ T cells from lung parenchyma to better understand which of the four populations are common or rare in the human lung.

2) The effect of different cytokines and chemokines on Mtb growth within the microspheres in Figure 5A-C and Figure 5—figure supplement 1 and their description in the text is confusing. Can the authors clarify what "Chemokines (RANTES, MIP-1a and MIP-1b) and growth factors (G-CSF and GM-CSF) had a small growth-promoting effect" means given that only GM-CSF led to significant increases in Mtb growth in Figure 5B? Additionally, some of the supplementary data in Figure 5—figure supplement 1 does not match up with the results shown in Figure 5. For example, Figure 5B shows no difference in Mtb growth with the addition of IL-1RA, but there is a significant increase in Mtb growth following IL-1RA treatment shown in Figure 5—figure supplement 1. Close reading of the figure legends shows that the concentration of cytokines/chemokines added in Figure 5—figure supplement 1 was 5 times higher, but there is no mention of this in the text or explanation of why. Please clarify why these two different experiments were performed and provide an interpretation of the differing results. Microsphere have disparate cells which are constituents of peripheral blood mononuclear cells. This might include T cells, myeloid cells, and matrix. Please discuss how these data compare to previous experiments performed on isolated Mtb infected cells. Specifically, other groups Hirsch et al., 1994 showed that TNF in Mtb infected macrophages interfered with Mtb replication. Other have shown how early macrophage events are interfered with my TBF blockers, that would allow Mtb replication. These data will be compared with Figure 5A. Why is the data so different? Sam Behar's lab has demonstrated that T-cell derived GMCSF diminishes Mtb replication (MBio 2017 Oct). This will be contrasted with data, Figure 5B. Please discuss the discrepancy.

3) The authors very nicely demonstrate that the increase in Mtb growth is dependent on TNF production and cite several papers that link TNF to Mtb growth by different mechanisms including "recruitment of excessive inflammatory cells and destruction of the extracellular matrix" and "cell survival." Is it possible to identify which mechanism is responsible for the Mtb growth phenotype in this 3D culture model or at least rule out any of these possibilities? For example, following PD-1 blockade with or without anti-TNF antibodies is there a change in the viability of myeloid cells or a difference in the proportions of cell populations within the microspheres? How do the authors envision the mechanism in this system might be similar or different from other PD-1 inhibition studies? For instance, there is obviously no inflammatory cell recruitment occurring within the microspheres but perhaps TNF production leads to changes in population frequencies that impact bacterial control. The study would be strengthened by an analysis to determine which cell type or mechanism is responsible for the excess TNF production following PD-1 blockade (CD4^+^ T cells, CD8^+^ T cells, or an innate cell type). At minimum, the authors should include a nuanced which of the mechanisms would be relevant to their model and why.

---

## [Author Response]

Essential revisions:1) Figure 1: PD-1 is expressed in granulomas but not in areas of immunopathology. This is an important point being made, with human samples, but the first statistical testing is PD1 levels in tissue cells versus blood cells. Perhaps a) cells become PD1 positive in tissue, b) PD1 cells hone to the tissue, c) other? Though p-values are reported, it wasn't clear what the hypothesis was. It was much clearer in the tissue sections when the authors set out to determine whether the PD1+ were and were not, in the tissue. Maybe the statistical testing should be reserved for the data that directly address the experimental hypotheses?

We accept this comment. The point that we were trying to make was that PD-1 is expressed by T cells within the lung of patients with TB, and so we first show that PD-1 expressed in the lung, and this is at a higher level than in the blood. We have amended the statistical analysis in panel B as suggested to address the experimental hypothesis and to make the conclusions we are drawing clearer. We have also edited the text describing these analyses to make the hypothesis being tested apparent.

In Figure 1A, there are ~ 35 dots in the lung plot with a median frequency of PD1+ cells of about 15%. Does this mean that the authors studied 35 independent samples from 35 patients?

We have clarified the number in the text and legend to show that each dot represents a different patient, and there were 35 patients studied.

Please describe the N for the pathology samples in the Results (perhaps this was in a supplementary file?).

We have stated the N in the legend, which was 6 TB clinical biopsies.

It seems that the biologic difference between blood and lung is very thin and the statistical difference is driven by the large number of dots. It would be important to clarify the number of independent lung samples studied for this figure.

We have clarified the number in the legend, which was 35 samples.

Finally, in the text, the authors conclude by saying: "PD-1 is expressed by lung resident T-cells and may be required to prevent destructive lung disease." I believe this is based on an inverse correlation. It might be preferable to conclude the section with what is observed, then introduce the hypothesis in the next section.

We agree with this point and have amended the text as suggested, moving the hypothesis to the next section.

There are other critical questions that must be addressed in Figure 1. How does the frequency of PD-1 expressing T cells in TB patients compare to healthy controls? It would be helpful in Figure 1A to compare the frequencies (at least in the blood) to those in healthy controls to understand whether the increase in PD-1 is lung specific or systemic during Mtb infection.

PD-1 and PD-L1/2 expression has previously been reported to be elevated in circulating PBMCs in TB, in work we cited (Shen et al., 2016), and we now expand on this in the second paragraph of the Discussion. For the lung, the reviewer raises an important question. Unfortunately, out studies use lung tissue removed by surgical resection of patients with complications of TB, such as massive haemoptysis, and thus healthy control material is not easily obtainable to make these comparisons. Recently published work looking at healthy lungs from organ donors does report high PD-1 expression. This is now referenced and discussed in the first paragraph of the subsection “PD-1 is expressed in human TB granulomas but not in areas of immunopathology”.

Second, in Figure 1B, PD-1 expression appears to be highest on the CD103+ CD69+ population for CD8^+^ T cells, but CD103 does not seem to mark the CD4^+^ T population with the highest PD-1 expression. Is CD103 known to mark tissue residency for CD4^+^ T cells like it does for CD8^+^ T cells? If not, it might be better to leave this marker out of the analysis in Figure 1B-I for clarity.

Human lung tissue does contain CD69+CD103+ CD4 and CD8 T-cells. Although CD103+ expression in higher in the CD8^+^ fraction, approximately 20% of CD69+ve T-cells in healthy lung express CD103+, as published (Snyder et al., 2019). We clarify this point and cite this in the second paragraph of the subsection “PD-1 is expressed in human TB granulomas but not in areas of immunopathology”.

It would also be useful to include the overall frequencies of each of the four CD103/CD69 populations for both CD4^+^ and CD8^+^ T cells from lung parenchyma to better understand which of the four populations are common or rare in the human lung.

We now include these data as Supplementary file 1.

2) The effect of different cytokines and chemokines on Mtb growth within the microspheres in Figure 5A-C and Figure 5—figure supplement 1 and their description in the text is confusing. Can the authors clarify what "Chemokines (RANTES, MIP-1a and MIP-1b) and growth factors (G-CSF and GM-CSF) had a small growth-promoting effect" means given that only GM-CSF led to significant increases in Mtb growth in Figure 5B?

We apologise for the lack of clarity. The experiments were performed twice, with two concentrations of cytokines on each occasion. One concentration was that measured in the cell culture supernatants (Figure 4), and the second was 5x this, to allow for potential concentration gradients from the exterior to the centre of the microspheres. The 1x concentration (“low”) experiments are presented in the main manuscript, whilst the 5x concentration (“high”) experiments are in the supplementary material. The direction of effect of each cytokine was the same for low and high concentrations, though with a greater effect size for the 5x. We have provided clarification in –the first paragraph of the subsection “Exogenous TNF-α increases Mtb growth in microspheres” and Figure 5 and Figure 5—figure supplement 1 legend to avoid the confusion. We have also edited to state that only GM-CSF had a significant increase in addition to TNF-α, as the reviewer correctly states that we over-reported the minor chemokine effect.

Additionally, some the supplementary data in Figure 5—figure supplement 1 does not match up with the results shown in Figure 5. For example, Figure 5B shows no difference in Mtb growth with the addition of IL-1RA, but there is a significant increase in Mtb growth following IL-1RA treatment shown in Figure 5—figure supplement 1. Close reading of the figure legends shows that the concentration of cytokines/chemokines added in Figure 5—figure supplement 1 was 5 times higher, but there is no mention of this in the text or explanation of why. Please clarify why these two different experiments were performed and provide an interpretation of the differing results.

We addressed this critique above, and apologise that the manuscript was not sufficiently clear about the concentration differences between Figure 5 and Figure 5—figure supplement 1, and the rationale for using these two concentrations. We have further clarified in the first paragraph of the subsection “Exogenous TNF-α increases Mtb growth in microspheres” and in the respective figure legends.

Microsphere have disparate cells which are constituents of peripheral blood mononuclear cells. This might include T cells, myeloid cells, and matrix. Please discuss how these data compare to previous experiments performed on isolated Mtb infected cells. Specifically, other groups Hirsch et al., 1994 showed that TNF in Mtb infected macrophages interfered with Mtb replication. Other have shown how early macrophage events are interfered with my TBF blockers, that would allow Mtb replication. These data will be compared with Figure 5A. Why is the data so different? Sam Behar's lab has demonstrated that T-cell derived GMCSF diminishes Mtb replication (MBio 2017 Oct). This will be contrasted with data, Figure 5B. Please discuss the discrepancy.

We agree that the effect of individual cytokines is likely to be both concentration, context and host dependent, and that our results are at odds with those that show a protective effect of GM-CSF from the Behar laboratory that were performed in murine macrophages. We propose that for several cytokines there is a bell-shaped response curve, with an optimal concentration for mycobacterial control, and either very low or high concentrations being deleterious, as is emerging for TNF-α, from both our work and other groups. We accept that these concepts were not sufficiently commented on in the original submission, and have extended the Discussion (fourth paragraph), putting our results in the context of the previous work cited above, and we have expanded on the balance between protective and harmful concentrations of cytokines.

3) The authors very nicely demonstrate that the increase in Mtb growth is dependent on TNF production and cite several papers that link TNF to Mtb growth by different mechanisms including "recruitment of excessive inflammatory cells and destruction of the extracellular matrix" and "cell survival." Is it possible to identify which mechanism is responsible for the Mtb growth phenotype in this 3D culture model or at least rule out any of these possibilities? For example, following PD-1 blockade with or without anti-TNF antibodies is there a change in the viability of myeloid cells or a difference in the proportions of cell populations within the microspheres?

To address this comment, we have performed additional experiments studying macrophage polarisation with addition of TNF-α (Figure 5—figure supplement 3). High TNF concentrations lead to a reduction in CD80 expression, which associate with macrophages more permissive to Mtb growth. We now discuss this in the last paragraph of the subsection “Exogenous TNF-α increases Mtb growth in microspheres”. Further mechanistic dissection of the phenomenon will be possible within the model, which we propose to do by single cell sequencing of cells released from microspheres as discussed in the sixth paragraph of the Discussion. However, this work is beyond the scope of the current manuscript and clearly cannot be performed within the time frame available.

How do the authors envision the mechanism in this system might be similar or different from other PD-1 inhibition studies? For instance, there is obviously no inflammatory cell recruitment occurring within the microspheres but perhaps TNF production leads to changes in population frequencies that impact bacterial control.

We demonstrate a change in macrophage polarisation as above (Figure 5—figure supplement 3) but accept that our in vitro system cannot model the complexity of an in vivo system, where cellular recruitment will play a role. We discuss the benefits and limitations of the in vitro model in the sixth paragraph of the Discussion.

The study would be strengthened by an analysis to determine which cell type or mechanism is responsible for the excess TNF production following PD-1 blockade (CD4^+^ T cells, CD8^+^ T cells, or an innate cell type). At minimum, the authors should include a nuanced which of the mechanisms would be relevant to their model and why.

We completely agree that further detailed mechanistic study would strengthen the work, and we propose to address this by combining the microsphere system with single cell RNAseq, so that we can undertake detailed profiling of individual cellular phenotype within each condition. We can then determine phenotypic changes with PD-1 inhibition and how excess TNF leads to increased cell growth. However, this is an extensive programme of work, involving technical challenges of setting up single cell RNAseq in the containment level 3 laboratory, and cannot be performed within the time frame available. Therefore, to address the point, we have edited the manuscript as suggested to give a more nuanced discussion of the potential mechanism, and how they might be investigated, –in the sixth paragraph of the Discussion, as requested by the reviewers.